# Towards Practical Control of Singular Values of Convolutional Layers

**Alexandra Senderovich**[*][†]
HSE University

**Ekaterina Bulatova**[*]
HSE University

**Anton Obukhov**
ETH Zürich

**Maxim Rakhuba**
HSE University

## Abstract

In general, convolutional neural networks (CNNs) are easy to train, but their essential properties, such as generalization error and adversarial robustness, are hard to control. Recent research demonstrated that singular values of convolutional layers significantly affect such elusive properties and offered several methods for controlling them. Nevertheless, these methods present an intractable computational challenge or resort to coarse approximations. In this paper, we offer a principled approach to alleviating constraints of the prior art at the expense of an insignificant reduction in layer expressivity. Our method is based on the tensor-train decomposition; it retains control over the actual singular values of convolutional mappings while providing structurally sparse and hardware-friendly representation. We demonstrate the improved properties of modern CNNs with our method and analyze its impact on the model performance, calibration, and adversarial robustness. The source code is available at:
https://github.com/WhiteTeaDragon/practical_svd_conv

## 1   Introduction

Over the past decade, empirical advances in Deep Learning have made machine learning ubiquitous to researchers and practitioners from various fields of science and industry. However, the theory of Deep Learning lags behind its practical applications, resulting in unexpected outcomes or lack of explainability of the models. These adverse effects are becoming more prominent with many models put into customer-facing products, such as perception systems of self-driving cars, chatbots, and other applications. The demand for tighter control over deployed models has given rise to several subfields of Deep Learning, such as the study of adversarial robustness and model calibration, to name a few.

Convolutional neural networks (CNNs) and, in particular, residual CNNs have become a benchmark for various computer vision tasks. The key component of CNNs is convolutional layers, representing linear transformations that account for the image data structure. The singular values of these linear transformations are key to the properties of the whole network, such as the Lipschitz constant and, hence, generalization error and robustness to adversarial examples. Moreover, bounding the Lipschitz constant of convolutional layers can also improve training, as it prevents gradients from exploding.

Nevertheless, finding and controlling singular values of convolutional layers is challenging. Indeed, computing and then clipping exact singular values of a layer [Sedghi et al., 2019], given by a kernel tensor of the size $k \times k \times c_{\text{in}} \times c_{\text{out}}$, has time and space complexities of $\mathcal{O}(n^2 c^2(c + \log n))$ and $\mathcal{O}(n^2 c^2)$ respectively, where $n \times n$ is the input image size, $c_{\text{in}}$ and $c_{\text{out}}$ are the numbers of input and output channels respectively, $c = \max\{c_{\text{in}}, c_{\text{out}}\}$, and $k \times k$ is the filter size. Since typically $k^2 \ll c$, computing singular values requires many more operations than computing a single convolution with

---

[*]Equal contribution.
[†]Corresponding author: Alexandra Senderovich (AlexandraSenderovich@gmail.com)

36th Conference on Neural Information Processing Systems (NeurIPS 2022).

its asymptotic complexity $\mathcal{O}(n^2 c^2 k^2)$. When computing singular values, the storage of an arising kernel tensor of the size $n \times n \times c_{\text{in}} \times c_{\text{out}}$ (padded to image dimensions) can also be an issue for larger networks and inputs. This effect becomes even more pronounced in three-dimensional convolutions used for volumetric data processing. Alternative methods that parametrize the convolutional layers are also computationally demanding [Singla and Feizi, 2021b, Li et al., 2019].

In this paper, we propose a practical approach to constraining the singular values of the convolutional layers based on intractable techniques from the prior art. Our approach relies on the assumption that modern over-parameterized neural networks can be made sparse using tensor decompositions, incurring insignificant degradation of the downstream task performance. We investigate the impact of using our approach on model performance, calibration, and adversarial robustness. More specifically, our contributions are as follows:

1. We propose a new framework for reducing the computational complexity of controlling singular values of a convolutional layer by imposing the Tensor-Train (TT) decomposition constraint on a kernel tensor. It allows for substituting the computation of singular values of the original layer by a smaller layer. As a result, it reduces the complexity of singular values control by using a method of choice.

2. We extend the formula of exact singular values of a convolutional layer [Sedghi et al., 2019] to the case of strided convolutions, which is ubiquitous in CNN architectures.

3. We apply the proposed framework to different methods of controlling singular values and test them on several CNN architectures. Contrary to measuring just the downstream task performance as in the prior art, we additionally demonstrate improvements in adversarial robustness and model calibration of the networks.

The paper is organized as follows: Sec. 2 provides an overview of prior art on singular values and Lipschitz constant estimation, related methods employing these techniques, and assumptions used to develop our approach; Sec. 3 revisits computation of singular values of convolutional layers and describes our approach to dealing with computational complexity; Sec. 4 walks through the steps of our algorithm to computing singular values of compressed convolutional layers; Sec. 5 contains the empirical study of our method; Sec. 6 wraps up the paper.

## 2 Related Work

### 2.1 Computing Singular Values of Convolutions

Traditional discrete convolutional layers used in signal processing and computer vision [LeCun et al., 1989] can be seen from two perspectives. On the one hand, a $d$-dimensional convolution linearly maps (potentially overlapping) windows of a $c$-channel input signal with side $k$ into vectors of $c$ output values (*window map*). This perspective is related to the one implementation of convolutional layers, decomposing the operation into window extraction termed `im2col` [Chellapilla et al., 2006], followed by matrix multiplication via `GEMM` [Blackford et al., 2002]. From this perspective, the transformation is defined by the convolution weight tensor unfolded into a matrix of size $k^2 c_{\text{in}} \times c_{\text{out}}$ (`Conv2D` case); its singular values can be computed and controlled through SVD-based schemes over the said unfolding matrix. For example, Yoshida and Miyato [2017] proposed a method of Spectral Normalization to stabilize GAN training [Goodfellow et al., 2020]. Their method effectively estimates the largest singular value of the map via the power method. In a similar vein, Liu et al. [2019], Obukhov et al. [2021] explicitly parametrized the SVD decomposition of the mapping to control multiple singular values during training.

Nevertheless, the above perspective lacks the generality of treating the input signal as a whole, which is especially important when chaining maps, such as seen in deep CNNs. To this end, a discrete convolutional layer can be viewed as a mapping from and onto the space defined by the signal shape (*signal map*). The map is defined by an $N$-ly block-circulant matrix, where $N$ is the number of spatial dimensions of the convolution. A straightforward way to compute singular values of such a matrix by applying a full SVD algorithm is prohibitively expensive, even for small numbers of channels, as well as signal and window sizes. To overcome this issue, Sedghi et al. [2019] proposed a method based on the Fast Fourier Transform (FFT) for computing exact singular values of convolutional layers that has a much better complexity than the naive approach. Nevertheless, it is still quite demanding in space

and time complexities, as it deals with a padded kernel of the size $n \times n \times c \times c$. Alternatively, Singla and Feizi [2021b] directly impose nearly orthogonal constraints on convolutional layers using Taylor expansion of a matrix exponential of a skew-Hermitian matrix, which appears to be more efficient than a Block Convolution Orthogonal Parameterization approach of Li et al. [2019] proposed earlier. Singla and Feizi [2021a] drew connections between both perspectives and proved that the Lipschitz constant of the window map could serve as a genuine upper bound of the Lipschitz constant of the signal map if multiplied by a certain constant.

## 2.2 Effects of Singular Values

The study of singular values and the closely related Lipschitzness of convolutional neural networks impacts many domains of Deep Learning research. One essential expectation about neural networks is to generalize input data instead of memorizing it. Bartlett et al. [2017] present the generalization bound that depends on the product of the largest singular values of layers' Jacobians. Gouk et al. [2021] performed a detailed empirical study of the influence of bounding individual Lipschitz constants of each layer and its effect on the generalization error of the whole CNN. Singular values of Jacobians greater or smaller than $1.0$ can also be responsible for the growth or decay of gradients. Therefore, controlling all singular values can also be helpful to avoid exploding or vanishing gradients. The decay of singular values in layers also plays a role in the performance of GAN generators [Liu et al., 2019].

Apart from boosting the accuracy of predictions, researchers have been working on improving the robustness of neural networks to adversarial attacks, which is also affected by the Lipschitz constant [Cisse et al., 2017]. Adversarial attacks aim to find a negligible (to human perception) perturbation of input data that would sabotage the predictions of a model. In this paper, we use the AutoAttack Robust Accuracy module [Croce and Hein, 2020] to evaluate results. We additionally use a metric of classification model calibration (Expected Calibration Error, ECE) [Naeini et al., 2015]. Connections between model Lipschitz constraints, calibration, and out-of-distribution (OOD) model performance have been drawn in recent literature [Postels et al., 2022]. Various training techniques can also affect model properties [Kodryan et al., 2022] and efficiency of training [Gusak et al., 2022].

## 2.3 Tensor Decompositions in Deep Learning

Our proposed framework for CNN weights reparameterization relies on the well-studied property of neural network overparameterization [Frankle and Carbin, 2019]. Many network compression approaches [Anwar et al., 2017, Lee et al., 2019] agree that it is often possible to approach the uncompressed model performance by imposing some form of sparsity on the network weights. Tensor decompositions have been used previously for compression [Novikov et al., 2015, Garipov et al., 2016, Wang et al., 2018, Obukhov et al., 2020], multitask learning [Kanakis et al., 2020], neural rendering [Obukhov et al., 2022] and reinforcement learning [Sozykin et al., 2022]. The idea of this paper is to utilize tensor decomposition closely related to the SVD decomposition and to take advantage of its sparsity to reduce the complexity of controlling singular values. For that, we resort to the TT decomposition [Oseledets, 2011], which in the outlined context is also equivalent to the Tucker-2 decomposition [Kim et al., 2016]. As opposed to alternative tensor decompositions, this one allows us to naturally access singular values of the convolution layer with any method of choice.

## 3 Method-independent Complexity Reduction

This section explains how to reduce the complexity of calculating the singular values of a convolutional layer (in the *signal map* sense explained in Sec. 2.1), regardless of the chosen method. For the sake of generality, we consider layers acting on $(d+1)$-dimensional input arrays (tensors) with $d$ spatial dimensions and 1 channel dimension. Of particular interest are cases $d = 2$ and $d = 3$, corresponding to regular images and volumetric data.

We bootstrap our approach based on the recent research on neural network sparsity by compressing convolutional layers with the TT decomposition. We focus on dealing with the channel-wise complexity of the methods employed. In what follows, Sec. 3.1 revisits the convolutional operator in neural networks; Sec. 3.2 introduces our principled approach to compressing convolutional layers and reducing the complexity of the problem by a significant margin.

## 3.1 Regular Convolutional Layers

To introduce a convolutional layer formally, let $\mathcal{K} \in \mathbb{R}^{k \times \cdots \times k \times c_{\text{in}} \times c_{\text{out}}}$ be a $(d+2)$-dimensional kernel tensor, where $k$ is a filter size and $c_{\text{in}}, c_{\text{out}}$ are the numbers of input and output channels, respectively. A convolutional layer is given by a linear map $\mathscr{C}_{\mathcal{K}}$ – multichannel convolution with the kernel tensor $\mathcal{K}$ such that $\mathscr{C}_{\mathcal{K}} \colon \mathbb{R}^{c_{\text{in}} \times n \times \cdots \times n} \to \mathbb{R}^{c_{\text{out}} \times f(n,k) \times \cdots \times f(n,k)}$ and

$$(\mathscr{C}_{\mathcal{K}}(\mathcal{X}))_{j,:,\ldots,:} = \sum_{i=0}^{c_{\text{in}}-1} \mathcal{K}_{:,\ldots,:,i,j} * \mathcal{X}_{i,:,\ldots,:}, \quad j = 1, \ldots, c_{\text{out}}, \tag{1}$$

where $f(n,k)$ is an integer, depending on the type of the convolution $'*'$ used and convolution parameters, e.g., strides. The key assumption needed for our derivations is the bilinearity of $'*'$, which covers different convolution types, e.g., linear, periodic, and correlation. For example, a widely used correlation-type convolution with strides equal to one, reads

$$\mathcal{Y}_{jq_1 \ldots q_d} = \sum_{i=0}^{c_{\text{in}}-1} \sum_{p_1,\ldots,p_d=0}^{k-1} \mathcal{K}_{p_1 \ldots p_d ij} \mathcal{X}_{i,q_1+p_1,\ldots,q_d+p_d},$$

for all $j = 0, \ldots, c_{\text{out}} - 1$ and $q_\alpha = 0, \ldots, f(n,k) - 1$, $\alpha = 1, \ldots, d$, where $f(n,k) = n - k + 1$. In what follows in this section, we write

$$\mathcal{Y} = \mathscr{C}_{\mathcal{K}}(\mathcal{X}), \tag{2}$$

implying one of the convolution types mentioned above.

Using the linearity of $\mathscr{C}_{\mathcal{K}}$, we can rewrite (2) as a matrix-vector product:

$$\texttt{vec}(\mathcal{Y}) = T_{\mathcal{K}} \texttt{vec}(\mathcal{X}), \tag{3}$$

where $\texttt{vec}$ is a row-major reshaping of a multidimensional array into a column vector. In turn, $T_{\mathcal{K}} \in \mathbb{R}^{c_{\text{out}} n^d \times c_{\text{in}} n^d}$ is a matrix with an additional block structure: its $n^d \times n^d$ blocks are $d$-level Toeplitz matrices (see Sedghi et al. [2019] for $d = 2$). Representation (3) allows us to replace singular values of a linear map $\mathscr{C}_{\mathcal{K}}$ with singular values of a matrix $T_{\mathcal{K}}$, which is handy for analysis. In turn, access to singular values allows for controlling the Lipschitz constant of $\mathscr{C}_{\mathcal{K}}$ in terms of the largest singular value of $T_{\mathcal{K}}$, denoted as $\sigma_1(T_{\mathcal{K}})$. Indeed, since $\sigma_1(T_{\mathcal{K}}) = \|T_{\mathcal{K}}\|_2$, we have

$$\|\mathscr{C}_{\mathcal{K}}(\mathcal{X}) - \mathscr{C}_{\mathcal{K}}(\mathcal{Z})\|_F = \|T_{\mathcal{K}}(\texttt{vec}(\mathcal{X}) - \texttt{vec}(\mathcal{Z}))\|_2 \leq \|T_{\mathcal{K}}\|_2 \|\mathcal{X} - \mathcal{Z}\|_F = \sigma_1(T_{\mathcal{K}}) \|\mathcal{X} - \mathcal{Z}\|_F.$$

## 3.2 Compressed Convolutional Layers

Recall that even if $d = 2$, finding the SVD of $T_{\mathcal{K}}$ requires $\mathcal{O}(n^2(c^2 + \log n))$ FLOPs, which is too much for practical use with large networks. To reduce the computational cost of SVD of $T_{\mathcal{K}}$, we propose a low-rank compressed layer representation based on the following tensor decomposition:

$$\mathcal{K}_{p_1 \ldots p_d ij} = \sum_{\alpha=0}^{r_1-1} \sum_{\beta=0}^{r_2-1} \mathcal{K}^{(1)}_{i\alpha} \, \mathcal{K}^{(2)}_{p_1 \ldots p_d \alpha \beta} \, \mathcal{K}^{(3)}_{\beta j}, \tag{4}$$

where $\mathcal{K}^{(1)} \in \mathbb{R}^{c_{\text{in}} \times r_1}$, $\mathcal{K}^{(3)} \in \mathbb{R}^{r_2 \times c_{\text{out}}}$ and $\mathcal{K}^{(2)} \in \mathbb{R}^{k \times \cdots \times k \times r_1 \times r_2}$ are some small tensors and integers $r_1, r_2$: $1 \leq r_1 \leq c_{\text{in}}$, $1 \leq r_2 \leq c_{\text{out}}$ are called ranks. This decomposition is essentially the TT decomposition [Oseledets, 2011] of the kernel tensor with filter modes $p_1, \ldots, p_d$ merged into a single index. We also note that it is equivalent to the so-called Tucker-2 decomposition, successfully used by Kim et al. [2016] to compress large convolutional networks. The proposed decomposition is visualized in Fig. 1. We note that we chose the TT decomposition as it gives us access to the SVD decomposition of the convolution (Theorem 1). It is, however, unclear how to obtain similar results with other decompositions, such as tensor-ring or canonical tensor decomposition, as they do not admit SVD-like form, see, e.g., [Grasedyck et al., 2013] for more details of these formats.

By substituting (4) into (1), using bilinearity of $'*'$ and omitting summation limits for clarity:

$$(\mathscr{C}_{\mathcal{K}}(\mathcal{X}))_{j,:,\ldots,:} = \sum_i \mathcal{K}_{:,\ldots,:,i,j} * \mathcal{X}_{i,:,\ldots,:} = \sum_i \left( \sum_{\alpha,\beta} \mathcal{K}^{(1)}_{i\alpha} \, \mathcal{K}^{(2)}_{:,\ldots,:,\alpha,\beta} \, \mathcal{K}^{(3)}_{\beta j} \right) * \mathcal{X}_{i,:,\ldots,:} =$$

$$\sum_i \sum_{\alpha,\beta} \mathcal{K}^{(1)}_{i\alpha} \mathcal{K}^{(3)}_{\beta j} \left( \mathcal{K}^{(2)}_{:,\ldots,:,\alpha,\beta} * \mathcal{X}_{i,:,\ldots,:} \right) = \sum_\beta \mathcal{K}^{(3)}_{\beta j} \sum_\alpha \mathcal{K}^{(2)}_{:,\ldots,:,\alpha,\beta} * \left( \sum_i \mathcal{K}^{(1)}_{i\alpha} \mathcal{X}_{i,:,\ldots,:} \right),$$

we obtain that the convolution with kernel given by (4) is equivalent to a sequence of three convolutions without nonlinear activation functions between them:

1. $1 \times \cdots \times 1$ convolution with $c_{\text{in}}$ input and $r_1$ output channels (kernel tensor – $\mathcal{K}^{(1)}$);

2. $k \times \cdots \times k$ convolution with $r_1$ input and $r_2$ output channels (kernel tensor – $\mathcal{K}^{(2)}$);

3. $1 \times \cdots \times 1$ convolution with $r_2$ input and $c_{\text{out}}$ output channels (kernel tensor – $\mathcal{K}^{(3)}$);

or equivalently,

$$\mathscr{C}_{\mathcal{K}} = \mathscr{T}_{\mathcal{K}^{(3)}} \circ \mathscr{C}_{\mathcal{K}^{(2)}} \circ \mathscr{T}_{\mathcal{K}^{(1)}}, \tag{5}$$

where $\mathscr{A} \circ \mathscr{B}$ denotes the composition of two maps $\mathscr{A}$ and $\mathscr{B}$; $\mathscr{T}_{\mathcal{K}^{(i)}}$ denotes $1 \times \cdots \times 1$ convolution map given by a convolution kernel $\mathcal{K}^{(i)}$.

In the rest of this section, we show that singular values of the original convolutional layer with kernel $\mathcal{K}$ correspond to singular values of the compressed kernel $\mathcal{K}^{(2)}$. As a first step, we demonstrate that we can impose orthogonality constraints on $\mathcal{K}^{(1)}$ and $\mathcal{K}^{(3)}$ without loss of expressivity.

**Lemma 1.** *Let* $\mathcal{K} \in \mathbb{R}^{k \times \cdots \times k \times c_{\text{in}} \times c_{\text{out}}}$ *be given by* (4). *Then there exist matrices* $\mathcal{Q}^{(1)} \in \mathbb{R}^{c_{\text{in}} \times r_1}$, $\mathcal{Q}^{(3)} \in \mathbb{R}^{r_2 \times c_{\text{out}}}$ *satisfying* $\mathcal{Q}^{(1)\top} \mathcal{Q}^{(1)} = I_{r_1}$, $\mathcal{Q}^{(3)} \mathcal{Q}^{(3)\top} = I_{r_2}$ *and a tensor* $\mathcal{Q}^{(2)} \in \mathbb{R}^{k \times \cdots \times k \times r_1 \times r_2}$, *such that*

$$\mathcal{K}_{p_1 \ldots p_d i j} = \sum_{\alpha=0}^{r_1 - 1} \sum_{\beta=0}^{r_2 - 1} \mathcal{Q}^{(1)}_{i\alpha} \, \mathcal{Q}^{(2)}_{p_1 \ldots p_d \alpha \beta} \, \mathcal{Q}^{(3)}_{\beta j}. \tag{6}$$

*Proof.* See Sec. A.1 of the Appendix. □

Now we have all preliminaries in place to formulate the key result that our framework is based on.

**Theorem 1.** *Let* $\mathcal{K} \in \mathbb{R}^{k \times \cdots \times k \times c_{\text{in}} \times c_{\text{out}}}$ *be given by* (4). *Let also* $\mathcal{K}^{(1)} \in \mathbb{R}^{c_{\text{in}} \times r_1}$, $\mathcal{K}^{(3)} \in \mathbb{R}^{r_2 \times c_{\text{out}}}$:

$$\mathcal{K}^{(1)\top} \mathcal{K}^{(1)} = I_{r_1}, \quad \mathcal{K}^{(3)} \mathcal{K}^{(3)\top} = I_{r_2}. \tag{7}$$

*Then the multiset of nonzero singular values of* $\mathscr{C}_{\mathcal{K}}$ *defined in* (1) *equals the multiset of nonzero singular values of* $\mathscr{C}_{\mathcal{K}^{(2)}}$.

*Proof.* See Sec. A.2 of the Appendix. □

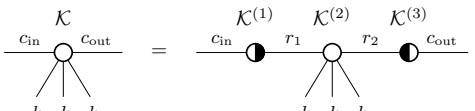

Figure 1: Tensor diagram representation of (4) for $d = 3$ with the constraints (7) as in Theorem 1. A node with $D$ "legs" represents a $D$-dimensional tensor; individual legs represent indices; connected legs represent summation over the respective indices; half-filled nodes are orthogonal matrices.

Theorem 1 suggests that after forcing the orthogonality of $\mathcal{K}^{(1)}$ and $\mathcal{K}^{(3)}$, one only needs to find singular values of the layer corresponding to the smaller kernel tensor $\mathcal{K}^{(2)}$ of the size $n \times n \times r_1 \times r_2$ instead of the original kernel tensor $\mathcal{K}$ of the size $n \times n \times r_1 \times r_2$. At the same time, thanks to Lemma 1, one may always set $\mathcal{K}^{(1)}$ and $\mathcal{K}^{(3)}$ to be orthogonal without additionally losing the expressivity of $\mathscr{C}_{\mathcal{K}}$. Fig. 1 demonstrates the resulting decomposition using tensor diagram notation.

## 4 Application of the Proposed Framework

This section discusses how to apply the proposed framework to various methods. We start with the explicit formula for singular values computation and then discuss how to apply the framework when the convolutions are parametrized, for example, as in [Singla and Feizi, 2021b, Li et al., 2019].

---

**Algorithm 1** Singular values of a TT-compressed periodic convolutional layer ($d = 2$).

---

**Input:**
  $\mathcal{K}^{(1)} \in \mathbb{R}^{c_{\text{in}} \times r_1}$ – left core tensor of the TT-compressed kernel $\mathcal{K}$,
  $\mathcal{K}^{(2)} \in \mathbb{R}^{k \times k \times r_1 \times r_2}$ – middle core tensor of the TT-compressed kernel $\mathcal{K}$,
  $\mathcal{K}^{(3)} \in \mathbb{R}^{r_2 \times c_{\text{in}}}$ – right core tensor of the TT-compressed kernel $\mathcal{K}$,
  $n$ – input image size,
  $s$ – stride of the convolution defined by kernel $\mathcal{K}$ s.t. $n \equiv 0 \pmod{s}$.

**Output:**
  Non-zero singular values of the convolutional layer, defined by the kernel tensor $\mathcal{K}$.

  1: $Q^{(1)}, R^{(1)} \leftarrow \mathtt{QR}\left(\mathcal{K}^{(1)}\right)$           ▷ Perform QR decomposition of the left core

  2: $Q^{(3)}, R^{(3)} \leftarrow \mathtt{QR}\left(\mathcal{K}^{(3)\top}\right)$         ▷ Perform QR decomposition of the right core

  3: $\widehat{\mathcal{K}}^{(2)}_{p_1,p_2,:,:} \leftarrow R^{(1)}\mathcal{K}^{(2)}_{p_1,p_2,:,:}R^{(3)\top}, \; \forall p_1, p_2$  ▷ Absorb non-orthogonal factors in the middle core

  4: $\widehat{\mathcal{K}}^{(2)} \leftarrow \mathtt{pad\_zeros}\left(\widehat{\mathcal{K}}^{(2)}, \; \mathtt{shape}{=}(n \times n \times r_1 \times r_2)\right)$   ▷ Pad middle core to the image size

  5: $R \leftarrow \mathtt{reshape}\left(\widehat{\mathcal{K}}^{(2)}, \; \mathtt{shape}{=}(8)\right)$       ▷ Reshape middle core as in (8). See also Sec. B.1.

  6: $P^{(:,:)}_{ij} \leftarrow \mathtt{fft2}\left(R_{j \bmod s^2,:,:,i,\lfloor j/s^2 \rfloor}\right), \; \forall \, i,j$     ▷ Perform 2-dimensional FFT of slices of $R$

  7: **return** $\displaystyle\bigcup_{p_1,p_2 \in \{1,\dots,\frac{n}{s}\}} \sigma\left(P^{(p_1,p_2)}\right)$     ▷ Return a union of all singular values of $P^{(\cdot,\cdot)}$

---

## 4.1 Explicit Formulas for Strided Convolutions

Sedghi et al. [2019] consider periodic convolutions, allowing them to express singular values through discrete Fourier transforms and several SVDs. We extend the main result of their work to the case of strided convolutions and summarize it in the following theorem.

**Theorem 2.** *Let $\widehat{\mathcal{K}} \in \mathbb{R}^{n \times n \times c_{\text{in}} \times c_{\text{out}}}$ be a kernel $\mathcal{K} \in \mathbb{R}^{k \times k \times c_{\text{in}} \times c_{\text{out}}}$ of a periodic convolution, padded with zeros along the filter modes. Let $s$ be a stride of the convolution $\mathscr{C}_{\mathcal{K}}$: $n \equiv 0 \pmod{s}$ and let the reshaped kernel $R \in \mathbb{R}^{s^2 \times \frac{n}{s} \times \frac{n}{s} \times c_{\text{in}} \times c_{\text{out}}}$ be such that:*

$$R_{q,a,b,i,j} = \widehat{\mathcal{K}}_{\lfloor q/s \rfloor + as, (q \bmod s) + bs, i, j}. \tag{8}$$

*Let us consider $\left(\frac{n}{s}\right)^2$ matrices $P^{(p_1,p_2)} \in \mathbb{R}^{c_{\text{in}} \times s^2 c_{\text{out}}}$ with entries:*

$$P^{(p_1,p_2)}_{ij} = (F^\top R_{j \bmod s^2,:,:,i,\lfloor j/s^2 \rfloor} F)_{p_1 p_2}, \tag{9}$$

*where $F$ is an $n \times n$ Fourier matrix. Then the multiset of singular values of $T_{\mathcal{K}}$ is as follows:*

$$\sigma\left(T_{\mathcal{K}}\right) = \bigcup_{p_1,p_2 \in \{1,\dots,\frac{n}{s}\}} \sigma\left(P^{(p_1,p_2)}\right). \tag{10}$$

*Proof.* See Sec. B.2 of the Appendix. $\qquad\square$

The algorithm for finding the singular values of a TT-compressed convolutional layer based on Theorem 2 is summarized in Alg. 1. This algorithm consists of several major parts. We start with a layer with the imposed TT structure as in (4). We reduce this decomposition to the form with orthogonality conditions (7) by using the QR decomposition (steps 1–3). The second part is the application of Theorem 2 to the smaller kernel tensor $\widehat{\mathcal{K}}^{(2)}$ (steps 4–7). For detailed pseudocode of applying Theorem 2, see Sec. B.1.

Computing singular values using (9) and (10) has the complexity $\mathcal{O}(n^2 c^2 (cs^2 + \log \frac{n}{s}))$, $c = \max\{c_{\text{in}}, c_{\text{out}}\}$, which depends cubically on $c$. Thus, reducing $c$ can significantly reduce the computational cost of finding the singular values. Let us now use the TT-representation (4) for kernel tensor with the ranks $r_1 = r_2 = r$. Theorem 1 suggests that we only need to apply Theorem 2 to the core $\mathcal{K}^{(2)} \in \mathbb{R}^{k \times k \times r_1 \times r_2}$, which leads to the complexity $\mathcal{O}(n^2 r^2 (r + \log \frac{n}{s}))$. For example, $r = c/2$ provides a theoretical speedup of up to 8 times (see Fig. 2 for practical evaluation). Memory consumption can also be high with the method of Sedghi et al. [2019]. Processing the padded kernel tensor of the size $n \times n \times c \times c$ with our method requires 4 times less memory for $r = c/2$ and 9 times less for $r = c/3$ (see also Fig. 2).

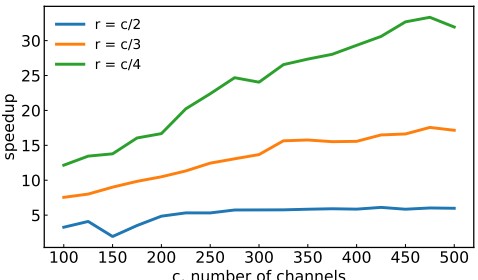 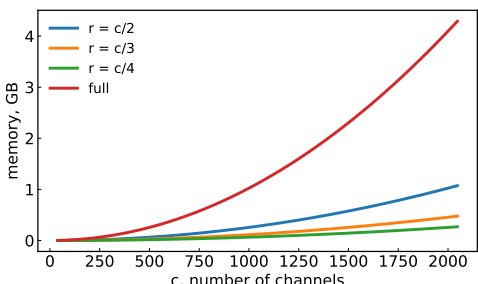

Figure 2: Left: the speedup of computing singular values of a TT-compressed layer using Alg. 1 relative to an uncompressed one. Right: Memory (single precision) to store the full $n \times n \times c \times c$ kernel and its TT-compressed form. Hyperparameter values: $n = 16$ and $s = 1$.

## 4.2 Orthogonality Constraints

To ensure that the frame matrices $\mathcal{K}^{(1)}$ and $\mathcal{K}^{(3)}$ are orthogonal, we apply QR decomposition to the kernel in Alg. 1. However, we can use other methods to maintain the orthogonality of frame matrices; for example, we can impose regularization on the layers. Regularization is quite helpful in case $\mathcal{K}^{(2)}$ is not a standard convolution but has a special complex structure, such as the SOC layer from [Singla and Feizi, 2021b]. For that, let $n_l$ denote the number of TT-decomposed layers in a network, $\mathcal{K}_i^{(1)}$ and $\mathcal{K}_i^{(3)}$ – left and right TT-cores of the $i^{\text{th}}$ layer, and $r_{1,i}, r_{2,i}$ – layer ranks. Then, the orthogonal loss reads as:

$$\text{loss}_{\text{ort}} = \left( \sum_{i=0}^{n_l} \left\| \mathcal{K}_i^{(1)\top} \mathcal{K}_i^{(1)} - I_{r_{1,i}} \right\|_F^2 + \left\| \mathcal{K}_i^{(3)} \mathcal{K}_i^{(3)\top} - I_{r_{2,i}} \right\|_F^2 \right) \Bigg/ \left( \sum_{i=0}^{n_l} r_{1,i}^2 + r_{2,i}^2 \right). \quad (11)$$

In the case of applying this regularization, we sum this loss with cross-entropy:

$$\text{loss} = \text{loss}_{\text{CE}} + \lambda_{\text{ort}} \text{loss}_{\text{ort}}. \quad (12)$$

The value of $\text{loss}_{\text{ort}}$ is the MSE measure of non-orthogonality of all $\mathcal{K}_i^{(1)}$ and $\mathcal{K}_i^{(3)}$, so the coefficient $\lambda_{\text{ort}}$ tends to be large. We set it to $1e5$ for experiments with WideResNets.

## 5 Experiments

To study the effects of the proposed regularization on the performance of CNNs, we combined it with various methods of controlling singular values and applied it during training on the image classification task on the CIFAR-10 and CIFAR-100 datasets [Krizhevsky et al., 2009]. The code was implemented in PyTorch [Paszke et al., 2019] and executed on a NVIDIA Tesla V100, 32GB. In our experiments, we used the LipConvNet architecture [Singla and Feizi, 2021b], a WideResNet [Zagoruyko and Komodakis, 2016], and VGG-19 [Simonyan and Zisserman, 2015]. Furthermore, the ranks are tested in the range $[c/4, c/2]$, where $c$ is the largest number of channels among all convolutions (except for $1 \times 1$), which we found to give the better performance/speedup trade-off.

**Evaluation Metrics**  We consider four main metrics: standard accuracy on the test split, accuracy on the CIFAR-C dataset (CC) [Hendrycks and Dietterich, 2019][1], accuracy after applying the AutoAttack module (AA) [Croce and Hein, 2020], and the Expected Calibration Error (ECE) [Guo et al., 2017]. The last columns of tables with the results of our experiments contain compression ratios of all layers in the network. In addition, we reported a dedicated metric for the LipConvNet architecture: p.r. – provable bound-norm robustness, introduced by Li et al. [2019]. This metric shows the percentage of input images guaranteed to be predicted correctly despite any perturbation within radius $36/255$.

Table 1: Comparison between SOC [Singla and Feizi, 2021b] and SOC combined with the proposed framework using LipConvNet-$N$ on CIFAR-10. "Speedup" is the speedup of training w.r.t. the SOC baseline. "Comp." (compression) is the ratio between the number of parameters of convolutional layers in the original and the decomposed networks. Despite speedups not being as substantial as in other experiments, we observe consistent improvement in all metrics, which is discussed in Fig. 3.

| RANK | $N$ | ACC. ↑ | AA ↑ | CC ↑ | ECE ↓ | P.R. ↑ | SPEEDUP↑ | COMP.↑ |
|------|-----|--------|------|------|-------|--------|----------|--------|
| – | 5 | 75.6 ± 0.3 | 31.1 ± 0.2 | 67.3 ± 0.1 | 8.6 ± 0.4 | 59.1 ± 0.1 | 1.0 | 1.0 |
| 128 | 5 | 76.9 ± 0.2 | 32.8 ± 0.5 | 68.8 ± 0.2 | 6.7 ± 0.1 | 62.7 ± 0.1 | 1.4 | 2.9 |
| 256 | 5 | **78.3 ± 0.1** | **34.7 ± 0.6** | **69.7 ± 0.2** | **5.3 ± 0.2** | **65.4 ± 0.3** | 0.9 | 1.2 |
| – | 20 | 76.3 ± 0.5 | 33.4 ± 0.5 | 68.1 ± 0.2 | 7.5 ± 0.1 | 61.4 ± 0.2 | 1.0 | 1.0 |
| 128 | 20 | 76.8 ± 0.2 | 33.0 ± 0.1 | 68.3 ± 0.1 | 5.9 ± 0.2 | 62.4 ± 0.2 | 1.2 | 3.3 |
| 256 | 20 | **78.4 ± 0.2** | **35.4 ± 0.5** | **70.4 ± 0.1** | **4.7 ± 0.4** | **65.6 ± 0.2** | 1.1 | 1.4 |
| – | 30 | 76.3 ± 1.0 | 32.0 ± 1.6 | 67.9 ± 1.0 | 7.3 ± 1.0 | 61.9 ± 1.2 | 1.0 | 1.0 |
| 128 | 30 | 76.0 ± 0.1 | 32.6 ± 0.1 | 68.0 ± 0.1 | 5.0 ± 0.6 | 61.7 ± 0.6 | 1.3 | 3.5 |
| 256 | 30 | **77.9 ± 0.1** | **34.7 ± 0.3** | **69.7 ± 0.3** | **4.8 ± 0.1** | **64.8 ± 0.2** | 1.1 | 1.5 |

## 5.1 Experiments on LipConvNet

Singla and Feizi [2021b] introduced a new architecture called Lipschitz Convolutional Networks, or LipConvNet for short. LipConvNet-$N$ consists of $N$ convolutional layers and activations. This architecture is provably 1-Lipschitz by contrast to popular residual network architectures. In Singla and Feizi [2021b], the authors also propose an orthogonal layer called Skew Orthogonal Convolution (SOC) and applied it in LipConvNet.

We modify the SOC layer in correspondence with the proposed framework by adding a $1 \times 1 \times c_{\text{in}} \times r$ convolution before applying it, and finally applying the last convolution of size $1 \times 1 \times r \times c_{\text{out}}$, thus reducing the number of channels to $r \times r$ in the bottleneck. To maintain the orthogonality of this new layer that we call SOC-TT, we add orthogonal loss (Sec. 4.2) to keep these two frame matrices orthogonal.

Tab. 1 demonstrates metrics improvement after replacing SOC with SOC-TT. For rank 256, we performed a grid search of the $\lambda_{\text{ort}}$ coefficient. The training protocol is the same as in Singla and Feizi [2021b] except for the learning rate of LipConvNet-20 with the SOC-TT layer, which we set to 0.05 instead of 0.1. To investigate the observed increase in metrics, we present histograms of empirically estimated Lischitz constants by using adversarial attacks in Fig. 3 (see Sec. C.1) with optimally chosen $\lambda_{\text{ort}}$ from (12). We observed that increasing $\lambda_{\text{ort}}$ forces the frame matrices in the TT decomposition to be closer to orthogonal and, as a result, the histogram converges to that of the original SOC method. At the same time, decreasing $\lambda_{\text{ort}}$ may lead to networks with constants larger than 1 but with better metrics.

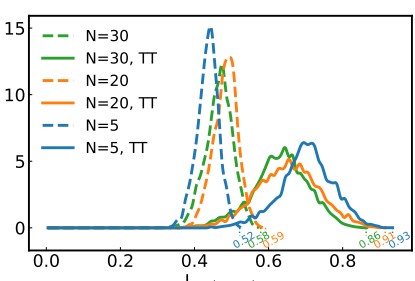

Figure 3: Histograms of empirically estimated Lipschitz constants of LipConvNet-$N$ architectures (see Sec. C.1). When applying the TT method to LipConvNet architecture, the peaks are wider and closer to 1, giving more flexibility to the model but still keeping constants below 1. Rank parameter: $r = 256$.

## 5.2 Experiments with WideResNet-16-10

Next, we consider WRN-16-10 [Zagoruyko and Komodakis, 2016] with almost 17M parameters. The results of our experiments on CIFAR-10 are summarised in Tab. 2. Additional experiments on CIFAR-100 are presented in Sec. C.3. The experimental setup and training schedule are taken from Gouk et al. [2021]. We train our models with batches of 128 elements for 200 epochs, using SGD optimizer and a learning rate of 0.1, multiplied by 0.1 after 60, 120, and 160 epochs. The hyperparameters for

[1]Distributed under Apache License 2.0

Table 2: Performance metrics for different constraints applied to WRN-16-10 trained on CIFAR-10. Both clipping and division combined with TT layers lead to an increase in metrics (except for ECE), with 256 being the best overall rank. Clipping tends to be a better option in terms of metrics but is substantially slower than the division approach. Moreover, clipping the singular values to 2 results in better performance than clipping to 1. "Speedup" is a speedup of the overhead resulting from singular values control in all the layers in the network. Clipping the whole network w/o decomposition takes 6.2 min while applying division – 0.6 sec. "Comp." (compression) is the ratio between the number of parameters of convolutional layers in the original ($\approx$ 16.8M) and decomposed networks.

| METHOD | RANK | ACC. ↑ | AA ↑ | CC ↑ | ECE ↓ | SPEEDUP ↑ | COMP. ↑ |
|---|---|---|---|---|---|---|---|
| | – | **95.52** | 54.22 | 72.91 | **2.06** | – | 1.0 |
| BASELINE | 192 | 95.21 | 54.97 | **73.94** | 2.84 | – | 3.6 |
| | 256 | 95.39 | **55.07** | 73.73 | 2.73 | – | 2.4 |
| | 320 | 95.02 | 52.32 | 72.86 | 2.85 | – | 1.8 |
| | – | **95.27** | 53.27 | **73.44** | 2.45 | 1.0 | 1.0 |
| CLIP TO 1 | 192 | 95.25 | 51.7 | 73.37 | 2.49 | 4.1 | 3.6 |
| | 256 | 95.12 | 51.94 | 72.47 | **2.43** | 3.3 | 2.4 |
| | 320 | 95.05 | **54.44** | 73.38 | 2.7 | 2.3 | 1.8 |
| | – | **95.99** | 55.77 | 73.27 | **1.92** | 1.0 | 1.0 |
| CLIP TO 2 | 192 | 95.45 | 55.45 | 73.26 | 2.55 | 4.1 | 3.6 |
| | 256 | 95.73 | **55.83** | 73.35 | 2.28 | 3.3 | 2.4 |
| | 320 | 95.5 | 54.63 | **74.34** | 2.53 | 2.3 | 1.8 |
| | – | 95.17 | 53.39 | 72.94 | 2.54 | 1.0 | 1.0 |
| DIVISION | 192 | **95.71** | 53.4 | **74.68** | **2.4** | 2.3 | 3.6 |
| | 256 | 95.7 | **56.28** | 73.43 | 2.51 | 1.5 | 2.4 |
| | 320 | 94.93 | 52.35 | 70.89 | 2.97 | 1.3 | 1.8 |

the experiments with division were taken from Gouk et al. [2021]. The orthogonal loss (11) was used in all experiments with decomposed networks.

**Clipping**   Following Sedghi et al. [2019], we apply the so-called clipping of singular values after computing the singular values of a layer using Theorem 2. In particular, we first fix a threshold parameter $\delta$, chosen to be 1 or 2 in numerical experiments. Then all singular values greater than $\delta$ are replaced with $\delta$. This procedure allows for maintaining maximal singular values at the desired level. After the clipping, we reconstruct the expanded kernel $\widehat{\mathcal{K}}_\delta$ of the size $n \times n \times c_{\text{in}} \times c_{\text{out}}$, which is no longer sparse. Finally, we revert to its original shape by setting $\mathcal{K} = \widehat{\mathcal{K}}_\delta(1{:}k, 1{:}k, :, :)$. This operation is performed every 100 training iterations; we provide the code for computing singular values in Sec. B.1 of the Appendix. The results suggest that applying TT decomposition not only compresses the network by decreasing the number of parameters of its convolutional layers, but also speeds up the clipping operation. Network robustness increased when training decomposed layers with clipping, which is confirmed by the results of the AutoAttack module.

**Division**   Even though we can bound the Lipschitz constants of all convolutional layers, the Lipschitz constant of the whole network can be arbitrary. In particular, for a WideResNet architecture, the Lipschitz constant would still be unconstrained due to residual connections, fully connected layers, batch normalization, and average pooling layers of the network. Gouk et al. [2021] derived a formula for the Lipschitz constants of batch normalization layers and proposed regularizing them too. After each training step, they set the desired values for Lipschitz constants of convolutional, fully connected, and batch normalization layers. To control the Lipschitz constant of convolutional and fully connected layers, they first compute the estimate of the largest singular value of a layer via power iteration. Then they divide the layer by the estimate and multiply it by the desired Lipschitz constant. A similar operation is performed for batch normalization layers, but the Lipschitz constant is computed directly.

In the original paper, this approach proved to be successful for WideResNet-16-10. The method itself already increases the robust metrics and accuracy substantially. The pattern remains intact after applying our decomposition to the layers: the robust accuracy of the AutoAttack module improves. Overall, the TT decomposition, applied without any other regularization, performs poorly. However,

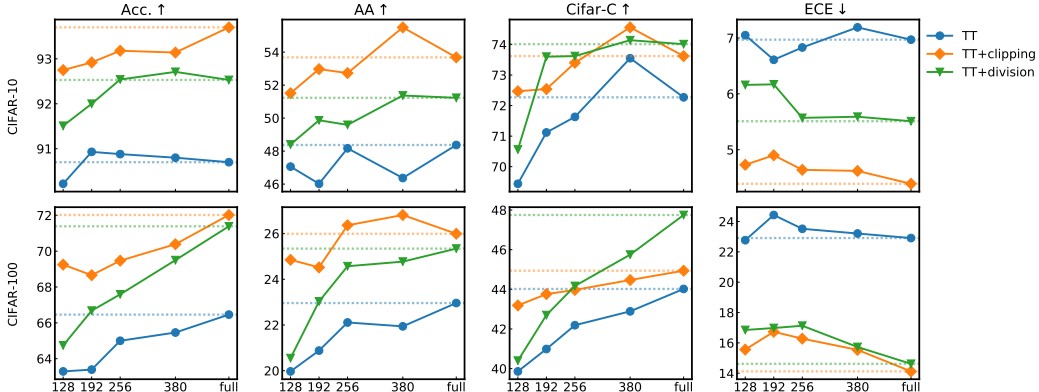

Figure 4: Metrics for different ranks on CIFAR-10 and CIFAR-100; "full" stands for the uncompressed case, i.e., "full" label with TT method is simply the baseline with no TT decomposition and no control of singular values, similarly "full" TT+clipping is [Sedghi et al., 2019] and "full" TT+division is [Gouk et al., 2021]. Horizontal dotted lines additionally represent uncompressed baselines.

when used to facilitate the application of a particular method of control over the singular values, the decomposition only slightly decreases the accuracy, which might still exceed that of the baseline.

## 5.3 Experiments with VGG-19

This section presents the results of controlling singular values on the VGG-19 network and two datasets: CIFAR-10 and CIFAR-100. As previously, we consider the clipping method with the exact computation of singular values [Sedghi et al., 2019] and the application of one iteration of the power method [Gouk et al., 2021]. The clipping parameter is set to $0.5$. The results are presented in Fig. 4. Firstly, we observe that all the considered robust metrics benefit from controlling singular values. This effect is more pronounced than for the WRN-16-10 and was also observed in [Gouk et al., 2021]. We conjecture that this happens due to the lack of residual connections in VGG-type architectures, which makes them more sensitive to singular values of convolutional layers; in residual networks, convolutional layers are only corrections to the signal. In almost all scenarios, we observed that the clipping method performed better than the division with the power method.

## 6 Conclusion

We proposed a principled sparsity-driven approach to controlling the singular values of convolutional layers. We investigated two different families of CNNs and analyzed their performance and robustness under the proposed singular values constraint. The limitations of our approach are partly inherited from the encapsulated methods of computing singular values (e.g., the usage of periodic convolutions) and partly stem from the core assumption of CNN layer sparsity. Overall, our approach has proven effective for large-scale networks with hundreds of input and output channels, making it stand out compared to intractable methods from the prior art.

**Societal Impact**   As our work is concerned with improving the robustness of neural networks, it positively impacts their reliability. Nevertheless, due to the generality of our approach, it can be used in various application domains, including malicious purposes.

## Acknowledgments

The publication was supported by the grant for research centers in the field of AI provided by the Analytical Center for the Government of the Russian Federation (ACRF) in accordance with the agreement on the provision of subsidies (identifier of the agreement 000000D730321P5Q0002) and the agreement with HSE University №70-2021-00139. The calculations were performed in part through the computational resources of HPC facilities at HSE University [Kostenetskiy et al., 2021].

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
