# Towards Practical Control of Singular Values of Convolutional Layers: Supplementary Materials

**Alexandra Senderovich**[*†]
HSE University

**Ekaterina Bulatova**[*]
HSE University

**Anton Obukhov**
ETH Zürich

**Maxim Rakhuba**
HSE University

## A  Compressed Convolutional Layers

### A.1  Proof of Lemma 1

The result essentially follows from different ways to represent the TT decomposition [Holtz et al., 2012]. By applying the QR decomposition to $\mathcal{K}^{(1)}$ and $\mathcal{K}^{(3)\top}$, we obtain:

$$\mathcal{K}^{(1)} = \mathcal{Q}^{(1)} R^{(1)}, \quad \mathcal{K}^{(3)\top} = \mathcal{Q}^{(3)\top} R^{(3)},$$

where $R^{(1)}$, $R^{(3)}$ are upper triangular. Substituting these formulas into (4) yields (6) with

$$\mathcal{Q}^{(2)}_{p_1 \ldots p_d \alpha \beta} = \sum_{\alpha'=0}^{r_1-1} \sum_{\beta'=0}^{r_2-1} R^{(1)}_{\alpha \alpha'} \mathcal{K}^{(2)}_{p_1 \ldots p_d \alpha' \beta'} R^{(3)}_{\beta' \beta},$$

which completes the proof.

### A.2  Proof of Theorem 1

Using (5), we have

$$T_{\mathcal{K}} = T_{\mathcal{K}^{(3)}} T_{\mathcal{K}^{(2)}} T_{\mathcal{K}^{(1)}}.$$

Let us first show that given $\mathcal{K}^{(1)\top} \mathcal{K}^{(1)} = I_{r_1}$, the matrix $T_{\mathcal{K}^{(1)}}$ has orthonormal rows, i.e., it satisfies

$$T_{\mathcal{K}^{(1)}} T_{\mathcal{K}^{(1)}}^{\top} = I_{r_1 n^d}. \tag{13}$$

To do so, let us find $T_{\mathcal{K}^{(1)}}$ in terms of $\mathcal{K}^{(1)}$. For any $\mathcal{X} \in \mathbb{R}^{c_{\text{in}} \times n \times \cdots \times n}$ and its row-major reshaping into a matrix $X \in \mathbb{R}^{c_{\text{in}} \times n^d}$, we have

$$\begin{aligned}
T_{\mathcal{K}^{(1)}} \text{vec}(\mathcal{X}) &\equiv \text{vec}\left(\mathscr{C}_{\mathcal{K}^{(1)}}(\mathcal{X})\right) \\
&= \text{vec}\left(\mathcal{K}^{(1)} X\right) = \left(\mathcal{K}^{(1)} \otimes I_{n^d}\right) \text{vec}(X) \\
&= \left(\mathcal{K}^{(1)} \otimes I_{n^d}\right) \text{vec}(\mathcal{X}).
\end{aligned}$$

Therefore, $T_{\mathcal{K}^{(1)}} = \mathcal{K}^{(1)} \otimes I_{n^d}$, where $\otimes$ denotes the Kronecker product of matrices. Using basic Kronecker product properties and the orthogonality of $\mathcal{K}^{(1)}$, we arrive at (13). Analogously, we may obtain $T_{\mathcal{K}^{(3)}} = \mathcal{K}^{(3)\top} \otimes I_{f(n,k)^d}$ and

$$T_{\mathcal{K}^{(3)}}^{\top} T_{\mathcal{K}^{(3)}} = I_{c_{\text{out}} f(n,k)^d}.$$

---

[*]Equal contribution.

[†]Corresponding author: Alexandra Senderovich (AlexandraSenderovich@gmail.com)

36th Conference on Neural Information Processing Systems (NeurIPS 2022).

Finally, using SVD of $T_{\mathcal{K}^{(2)}}$: $T_{\mathcal{K}^{(2)}} = U\Sigma V^\top$, we get:

$$T_{\mathcal{K}} = \left(\mathcal{K}^{(3)\top} \otimes I_{f(n,k)^d}\right) U\Sigma V^\top \left(\mathcal{K}^{(1)\top} \otimes I_{n^d}\right)$$

$$= \left(\left(\mathcal{K}^{(3)\top} \otimes I_{f(n,k)^d}\right) U\right) \Sigma \left(\left(\mathcal{K}^{(1)} \otimes I_{n^d}\right) V\right)^\top = \widetilde{U}\Sigma\widetilde{V}^\top,$$

where $\widetilde{U}, \widetilde{V}$ have orthonormal columns as a product of matrices with orthonormal columns. Hence, $T_{\mathcal{K}} = \widetilde{U}\Sigma\widetilde{V}^\top$ is in the compact SVD form, which completes the proof.

# B Periodic Strided Convolutions

## B.1 Code for Computing Singular Values of a Strided Convolution

According to Theorem 2, the following code computes the singular values of a transform encoded by a strided convolution. Note that neither the full SVD nor the clipping operation is included in the code. The code for computing the singular vectors and the new kernel with constrained singular values can be found in the source code repository.

```
def SingularValues(kernel, input_shape, stride):
 1   kernel_tr = np.transpose(kernel, axes=[2, 3, 0, 1])
 2   d1 = input_shape[0] - kernel_tr.shape[2]
 3   d2 = input_shape[1] - kernel_tr.shape[3]
 4   kernel_pad = np.pad(kernel_tr, ((0, 0), (0, 0), (0, d1), (0, d2)))
 5   str_shape = input_shape // stride
 6   r1, r2 = kernel_pad.shape[:2]
 7   transforms = np.zeros((r1, r2, stride**2, str_shape[0], str_shape[1]))
 8   for i in range(stride):
 9       for j in range(stride):
10           transforms[:, :, i*stride+j, :, :] = \
11                   kernel_pad[:, :, i::stride, j::stride]
12       transforms = np.fft.fft2(transforms)
13       transforms = transforms.reshape(r1, -1, str_shape[0], str_shape[1])
14       transpose_for_svd = np.transpose(transforms, axes=[2, 3, 0, 1])
15       sing_vals = svd(transpose_for_svd, compute_uv=False).flatten()
16       return sing_vals
```

## B.2 Proof of Theorem 2

In this section, we prove Theorem 2 from Sec. 4.1. Firstly, we analyze the structure of the matrix corresponding to a strided convolution. Secondly, we show that the columns of this matrix can be permuted to make matrix structure similar to that of a non-strided convolution. Therefore, the new theorem can be reduced to the already proven theorem.

Let us denote a circulant with each row shifted by the value of stride as a "strided circulant". The shape of such a strided circulant is $\frac{n}{s} \times n$, where $n$ is the number of elements in the first row. Here is an example of a strided circulant with $n = 4$, $s = 2$:

$$\begin{pmatrix} a & b & c & d \\ c & d & a & b \end{pmatrix}.$$

One can think of this strided circulant as a block-circulant matrix with block sizes $1 \times s$. At the same time, slicing this matrix by taking columns with a step equal to the stride, e.g., columns $\left(0, s, 2 \cdot s, \ldots \left(\frac{n}{s} - 1\right) \cdot s\right)$, gives a standard circulant matrix. This fact can be stated as

$$A = \sum_{i=0}^{s-1} A_i \left(I_{\frac{n}{s}} \otimes e_i^T\right),$$

where $\otimes$ denotes the Kronecker product, $A$ is a strided circulant, $e_i \in \mathbb{R}^s$ is the $i$-th standard basis vector in $\mathbb{R}^s$, and $A_i$ is a circulant obtained by slicing columns of $A$.

The regular 2D convolutional transform matrix with a single input and single output channels has a doubly block-circulant structure (see Section 5.5 in Jain [1989]). However, for strided convolutions, the structure is different. For a fixed pair of input and output channels, the output of a strided convolution is a submatrix of an output of a convolution with the same kernel but without the stride. More specifically, it is a slice with the stride $s$ by both dimensions (in Python, it would be written as `[::s, ::s]`). For the matrix encoding the transformation, it means that only every $s$-th block row (simulating the slice by the first dimension) and every $s$-th regular row of a block (simulating the slice by the second dimension) are considered. This means that the doubly block-circulant structure of the initial matrix turns into a doubly block-strided circulant structure of shape $\left(\frac{n}{s}\right)^2 \times n^2$, where $n$ is the size of a kernel.

Each block of this matrix is a strided circulant, and the block structure is that of a strided circulant as well. The matrix consists of $\frac{n}{s} \times n$ blocks, and each of them has the shape $\frac{n}{s} \times n$. If we denote a strided circulant of a row vector $a$ as $\mathrm{circ}_s(a)$, then the matrix of the transform is as follows:

$$
B = \begin{pmatrix}
\mathrm{circ}_s(\mathcal{K}_{0,:}) & \cdots & \mathrm{circ}_s(\mathcal{K}_{n-1,:}) \\
\mathrm{circ}_s(\mathcal{K}_{n-s,:}) & \cdots & \mathrm{circ}_s(\mathcal{K}_{n-s-1,:}) \\
\vdots & \ddots & \vdots \\
\mathrm{circ}_s(\mathcal{K}_{s,:}) & \cdots & \mathrm{circ}_s(\mathcal{K}_{s-1,:})
\end{pmatrix}.
$$

It can be noted that, in the same way as in strided circulants, we can take block columns of the matrix with a step equal to the stride and get block circulants (where each block is a strided circulant). This block structure can be described as follows:

$$
B = \sum_{i=0}^{s-1} B_i \left( \left( I_{\frac{n}{s}} \otimes e_i^T \right) \otimes I_n \right).
$$

Here $B$ is a doubly block-strided circulant matrix, and $B_i \in \mathbb{R}^{\left(\frac{n}{s}\right)^2 \times \frac{n^2}{s}}$ is a block-circulant matrix. This formula is similar to our previous sum representation for a strided circulant; however, the dimensions of the right term of each element are larger to account for the block structure of the left term. Note that this term is needed to describe how the columns of $B_i$ are positioned in the matrix $B$, similar to the 1D case.

Let us consider $B_i$. It consists of block columns $\left(i, s+i, 2s+i, \ldots \left(\frac{n}{s} - 1\right) s + i\right)$:

$$
B_i = \begin{pmatrix}
\mathrm{circ}_s(\mathcal{K}_{i,:}) & \cdots & \mathrm{circ}_s(\mathcal{K}_{n-s+i,:}) \\
\mathrm{circ}_s(\mathcal{K}_{n-s+i,:}) & \cdots & \mathrm{circ}_s(\mathcal{K}_{n-2s+i,:}) \\
\vdots & \ddots & \vdots \\
\mathrm{circ}_s(\mathcal{K}_{s+i,:}) & \cdots & \mathrm{circ}_s(\mathcal{K}_{i,:})
\end{pmatrix}
$$

As a block-circulant with $\frac{n}{s} \times \frac{n}{s}$ blocks of $\frac{n}{s} \times n$, it can be expanded as follows:

$$
B_i = \sum_{k=0}^{\frac{n}{s}-1} P^k \otimes C_{ik},
$$

where $C_{ik} \in \mathbb{R}^{\frac{n}{s} \times n}$ is a strided circulant block, $C_{ik} = \mathrm{circ}_s\left(\mathcal{K}_{i-sk,:}\right)$ and $P$ is a permutation matrix:

$$
P = \begin{pmatrix}
0 & \cdots & 0 & 1 \\
1 & \cdots & 0 & 0 \\
\vdots & \ddots & \vdots & \vdots \\
0 & \cdots & 1 & 0
\end{pmatrix} \in \mathbb{R}^{\frac{n}{s} \times \frac{n}{s}}.
$$

We can expand $C_{ik}$ as a strided circulant:

$$
C_{ik} = \sum_{j=0}^{s-1} A_{ikj} \left( I_{\frac{n}{s}} \otimes e_j^T \right),
$$

where $A_{ikj} \in \mathbb{R}^{\frac{n}{s} \times \frac{n}{s}}$ is a regular circulant matrix that can be acquired as $\mathrm{circ}_1(\mathcal{K}_{i-sk,j::s})$, i.e. as a circulant built from the slice of a string $\mathcal{K}_{i-sk}$, taken with a step $s$ starting from the index $j$. Finally,

$$B_i = \sum_{k=0}^{\frac{n}{s}-1} P^k \otimes C_{ik} = \sum_{k=0}^{\frac{n}{s}-1} P^k \otimes \left( \sum_{j=0}^{s-1} A_{ikj} \left( I_{\frac{n}{s}} \otimes e_j^T \right) \right) = \sum_{k=0}^{\frac{n}{s}-1} \sum_{j=0}^{s-1} P^k \otimes \left( A_{ikj} \left( I_{\frac{n}{s}} \otimes e_j^T \right) \right)$$

$$= \sum_{k=0}^{\frac{n}{s}-1} \sum_{j=0}^{s-1} P^k \otimes \left( A_{ikj} \otimes e_j^T \right) = \sum_{k=0}^{\frac{n}{s}-1} \sum_{j=0}^{s-1} \left( \left( P^k \otimes A_{ikj} \right) \otimes e_j^T \right) = \sum_{j=0}^{s-1} \left( \sum_{k=0}^{\frac{n}{s}-1} P^k \otimes A_{ikj} \right) \otimes e_j^T$$

$$= \sum_{j=0}^{s-1} \left( \sum_{k=0}^{\frac{n}{s}-1} P^k \otimes A_{ikj} \right) \left( I_{(\frac{n}{s})^2} \otimes e_j^T \right).$$

This reformulation helps us see that the slices of $B_i$ by columns with step $s$ are, in fact, doubly block-circulant matrices defined by spatial slices of the kernel. There are $s$ matrices $B_i$, each containing $s$ column slices. The $j$th column slice of $B_i$ is defined by $A_{i,:,j}$, which, in turn, is defined by the kernel slice $\mathcal{K}_{i::s,j::s}$.

In order to reduce the task of computing singular values of the matrix $B$, corresponding to the strided convolution, to the simpler task of computing singular values of a matrix corresponding to a regular convolution, let us permute the columns of $B_i$:

$$B_i' = \sum_{j=0}^{s-1} \left[ e_j^T \otimes \left( \sum_{k=0}^{\frac{n}{s}-1} P^k \otimes A_{ikj} \right) \right].$$

This matrix consists of $s$ consecutive doubly block-circulant matrices $\mathrm{circ}^2(\mathcal{K}_{i::s,j::s})$. The first dimension of $B_i$ is the same as the first dimension of $B$. Therefore, for $B$, this is also just a permutation of the columns. The next step is to permute the block columns of $B$:

$$B' = \sum_{i=0}^{s-1} \left( e_i^T \otimes \sum_{j=0}^{s-1} \left[ e_j^T \otimes \left( \sum_{k=0}^{\frac{n}{s}-1} P^k \otimes A_{ikj} \right) \right] \right).$$

After this permutation, matrix $B$ consists of $s^2$ consecutive doubly block-circulant matrices. Note that the particular order of these blocks is not important, as it is just a matter of column order.

Finally, let us look at the matrix associated with the multiple-channel convolution. The matrix of the convolutional transform with $c_{\mathrm{in}}$ input channels and $c_{\mathrm{out}}$ output channels is as follows (equivalent to the structure described in Sedghi et al. [2019]):

$$M = \begin{pmatrix} B_{0,0} & B_{0,1} & \cdots & B_{0,(c_{\mathrm{out}}-1)} \\ B_{1,0} & B_{1,1} & \cdots & B_{1,(c_{\mathrm{out}}-1)} \\ \vdots & \vdots & \ddots & \vdots \\ B_{(c_{\mathrm{in}}-1),0} & B_{(c_{\mathrm{in}}-1),1} & \cdots & B_{(c_{\mathrm{in}}-1),(c_{\mathrm{in}}-1)} \end{pmatrix},$$

where

$$B_{c,d} = \mathrm{circ}_s(\mathcal{K}_{:,:,c,d}).$$

However, now we know that we can permute the columns of this matrix to get a matrix comprised of doubly block-circulant blocks. After the permutation described above, we get a matrix

$$M' = \begin{pmatrix} B_{0,0}' & B_{0,1}' & \cdots & B_{0,(s^2 c_{\mathrm{out}}-1)}' \\ B_{1,0}' & B_{1,1}' & \cdots & B_{1,(s^2 c_{\mathrm{out}}-1)}' \\ \vdots & \vdots & \ddots & \vdots \\ B_{(c_{\mathrm{in}}-1),0}' & B_{(c_{\mathrm{in}}-1),1}' & \cdots & B_{(c_{\mathrm{in}}-1),(s^2 c_{\mathrm{out}}-1)}' \end{pmatrix},$$

where

$$B_{c,d}' = \mathrm{circ}^2 \left( \mathcal{K}_{i::s,j::s,c,\lfloor d/s^2 \rfloor} \right),$$

Table 3: Various metrics for the proposed framework applied to the SOC method (SOC-TT) and the LipConvNet-$N$ architectures. $\lambda_{\text{ort}}$ denotes the regularization parameter of the orthogonal loss (11). We chose a range of lambda values to allow us to keep the Lipschitz constant under 1. The rank is set to 256.

| PARAMETERS | | METRICS | | | | |
|---|---|---|---|---|---|---|
| -$N$ | $\lambda_{\text{ort}}$ | ACC. ↑ | CIFAR-C ↑ | ECE ↓ | AA ↑ | LIP. ↓ |
| | 5E3 | **78.37** | **69.84** | **5.42** | **34.65** | **0.93** |
| | 8E3 | 77.68 | 69.57 | 6.57 | 33.73 | 0.79 |
| 5 | 1E4 | 77.17 | 68.99 | 6.61 | 32.13 | 0.76 |
| | 3E4 | 76.25 | 67.9 | 8.35 | 30.96 | 0.6 |
| | 4E4 | 75.8 | 67.67 | 7.96 | 30.69 | 0.59 |
| | 5E4 | 75.74 | 67.66 | 8.71 | 30.77 | 0.58 |
| | 7E4 | **78.41** | **70.47** | **4.89** | **35.89** | **0.91** |
| | 8E4 | 78.34 | 70.27 | 5.18 | 35 | 0.89 |
| 20 | 1E5 | 77.45 | 69.81 | 5.43 | 34.1 | 0.8 |
| | 2E5 | 76.56 | 68.4 | 6.62 | 32.19 | 0.64 |
| | 3E5 | 76.03 | 68 | 7.23 | 32.52 | 0.58 |
| | 4E5 | 75.66 | 67.56 | 7.08 | 31.54 | 0.57 |
| | 2E5 | **77.82** | **69.94** | **4.78** | **34.91** | **0.86** |
| | 3E5 | 77.86 | 69.38 | 5.86 | 33.96 | 0.75 |
| 30 | 5E5 | 76.57 | 68.23 | 6.79 | 32.73 | 0.65 |
| | 7E5 | 75.93 | 68.03 | 6.93 | 32.65 | 0.59 |
| | 9E5 | 75.88 | 67.71 | 6.81 | 31.49 | 0.59 |
| | 1.2E6 | 76.22 | 67.6 | 7.23 | 31.85 | 0.56 |

$i, j$ – some integer indices from 0 to $s - 1$. The exact relationship between $i, j$ and $c, d$ is not important, as it depends only on the order of the columns. The only important thing is that it has to be the same in all the rows. We choose the following functional form:

$$i = \left\lfloor \left(d \mod s^2\right)/s \right\rfloor, \ \ j = d \mod s.$$

This matrix $M'$ can be perceived as the matrix of convolution for a new kernel $K' \in \mathbb{R}^{\frac{n}{s} \times \frac{n}{s} \times c_{\text{in}} \times c_{\text{out}} s^2}$, defined by this equation:

$$K_{a,b,c,d} = \widehat{\mathcal{K}}_{\lfloor (d \mod s^2)/s \rfloor + as, d \mod s + bs, c, d \mod s}.$$

Alternatively, to make things simpler, we can use the additional tensor $R$, defined in (8). It is easier to use this tensor for implementing the formula in Python.

To conclude, we reduced the task of computing the singular values of $M$ to the task of computing the singular values of the $M'$, solved by Sedghi et al. [2019]. The reduction is made possible via columns permutation, meaning that the singular values of the matrix did not change.

## C  Additional Empirical Studies

### C.1  Plotting Empirical Lipschitz Constant

In order to estimate the Lipschitz constants of LipConvNet networks and their TT-compressed counterparts, we plot histograms of empirical Lipschitz constants inspired by Sanyal et al. [2020]. The Lipschitz constants are evaluated by attacking each image from the test set using the FGSM attack of a fixed radius (0.5 in our experiments) [Goodfellow et al., 2015]. We compute a ratio that is upper-bounded by the true Lipschitz constant $L$:

$$L_{\text{estimated}}(\mathcal{X}) \equiv \frac{\|f(\mathcal{X}) - f(\mathcal{X}_{\text{attacked}})\|_2}{\|\mathcal{X} - \mathcal{X}_{\text{attacked}}\|_2} \leqslant L \tag{14}$$

for each image and plot the results as a histogram. Here, $f(\mathcal{X})$ is the output of a model $f$ for an input image $\mathcal{X}$; $\mathcal{X}_{\text{attacked}}$ is $\mathcal{X}$ perturbed, as described in the first paragraph. Then, by computing

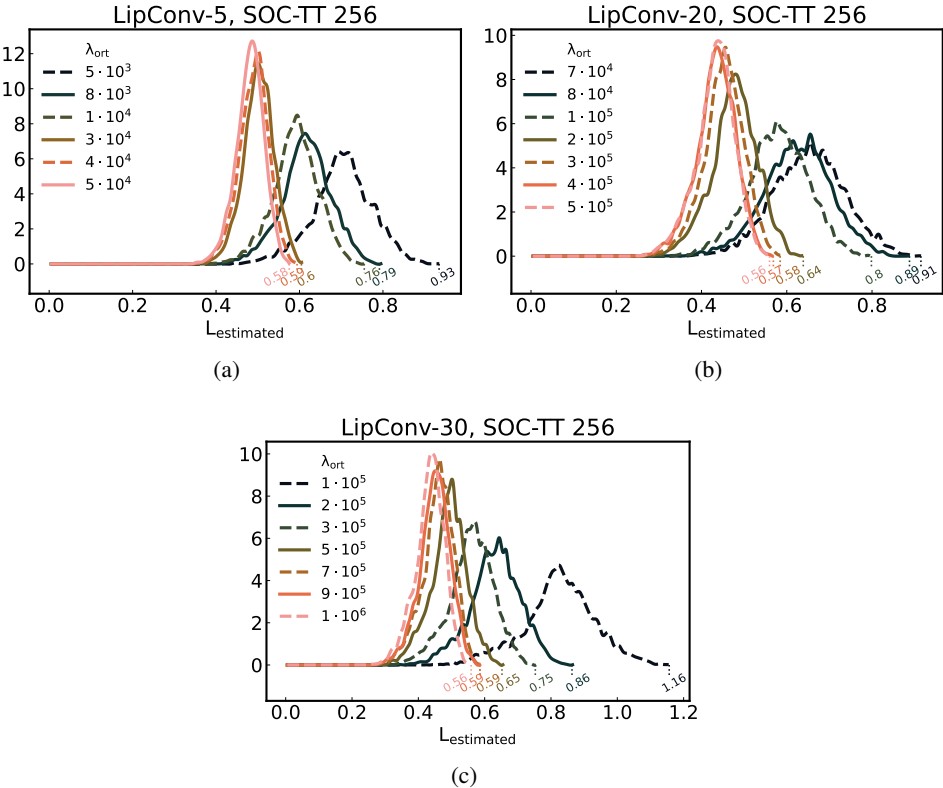

Figure 5: Histograms of empirical constants (14) of networks for different regularization parameters $\lambda_{\mathrm{ort}}$ and LipConvNet-$N$ architectures.

$L_{\mathrm{estimated}}(\mathcal{X})$ for different $\mathcal{X}$ from the dataset, we plot the histograms (Fig. 5). These histograms give us insights into the true Lipschitz constant of the network.

If all network layers are 1-Lipschitz, then we expect each $L_{\mathrm{estimated}}(\mathcal{X})$ on the histogram to be strictly bounded by 1. However, for the LipConvNet architectures, the bound on the Lipschitz constant might not be exact in practice due to truncating Taylor expansion. In our case, the frame matrices $\mathcal{K}^{(1)}$ and $\mathcal{K}^{(3)}$ are trained with the regularization loss, so they are not precisely orthogonal either. By increasing the coefficient $\lambda_{\mathrm{ort}}$ of this loss component, though, we can force the matrices to become orthogonal by the end of the training process.

To support this claim, we provide the distributions of the empirical Lipschitz constants for several LipConvNet architectures trained with different $\lambda_{\mathrm{ort}}$. The trained models are presented in Tab. 3. The last metric, "Lip.", or Maximum Empirical Lipschitz Constant, is a lower bound on the real Lipschitz constant of the corresponding model. It is obtained as the maximum value of all calculated empirical Lipschitz constants. As we can see, by increasing $\lambda_{\mathrm{ort}}$, we can balance between the 1-lipschitzness and the quality of metrics: the higher $\lambda_{\mathrm{ort}}$, the lower the metrics are and the more constrained the Lipschitz constant is. Numbers selected with bold correspond to models that were selected as best for the corresponding LipConvNet-$N$ architectures: they demonstrate high performance in terms of accuracy and robust metrics while at the same time maintaining the Maximum Empirical Lipschitz Constant that does not exceed 1. We also note that LipConvNet architectures do not converge if the convolutional layers are too far from being 1-Lipschitz. For example, this effect was observed on LipConvNet-5 with $\lambda_{\mathrm{ort}} = 4e3$, where the $\lambda_{\mathrm{ort}}$ turns out to be not big enough.

Fig. 5 demonstrates the shifts in distribution depending on $\lambda_{\mathrm{ort}}$. We can observe ranges of empirical Lipschitz constants on the $x$-axis. Maximum values for each model, denoted with dotted vertical lines, are the discussed Maximum Empirical Lipschitz Constants in Tab. 3.

## C.2 Inference Time of a SOC-TT Layer

Table 4: Performance metrics for different constraints applied to WideResNet-16-10 trained on CIFAR-100. Clipping to 1 increases the baseline performance only without TT decomposition, while the other methods (clipping to 2 and division) yield an increase in some of the metrics. "Speedup" is the speedup of an overhead resulting from singular values control in all the layers in the network. Clipping the whole network w/o decomposition takes 6.2 min, while the application of division takes 0.6 sec. "Comp." (compression) is the ratio between the number of parameters of convolutional layers in the original ($\sim 16.8$M) and decomposed networks.

| METHOD | RANK | ACC. ↑ | AA ↑ | CC ↑ | ECE ↓ | CLIP (S) ↓ | COMP. ↑ |
|---|---|---|---|---|---|---|---|
| BASELINE | – | 77.97 | **27.24** | **48.95** | **8.27** | – | 1.0 |
| | 192 | 78.53 | 25.5 | 47.28 | 11.78 | – | 3.6 |
| | 256 | **78.55** | 26.11 | 47.72 | 11.76 | – | 2.4 |
| | 320 | 76.51 | 24.81 | 46.18 | 12.5 | – | 1.8 |
| CLIP TO 1 | – | **78.99** | **27.84** | **47.89** | **10.09** | 1.0 | 1.0 |
| | 192 | 77.7 | 25.02 | 46.77 | 11.9 | 4.1 | 3.6 |
| | 256 | 77.71 | 27.15 | 46.24 | 11.88 | 3.3 | 2.4 |
| | 320 | 77.79 | 26.34 | 47.51 | 11.75 | 2.3 | 1.8 |
| CLIP TO 2 | – | **79.92** | **28.15** | **47.73** | **9.4** | 1.0 | 1.0 |
| | 192 | 78.74 | 27.21 | 46.08 | 11.41 | 4.1 | 3.6 |
| | 256 | 79.39 | 27.54 | 47.26 | 11.07 | 3.3 | 2.4 |
| | 320 | 79.52 | 26.82 | 47.68 | 10.43 | 2.3 | 1.8 |
| DIVISION | – | 78.59 | **26.85** | **48.38** | 10.97 | 1.0 | 1.0 |
| | 192 | 78.65 | 25.4 | 47.86 | 8.29 | 4.1 | 3.6 |
| | 256 | **78.98** | 26.26 | 47.77 | **8.21** | 3.3 | 2.4 |
| | 320 | 77.58 | 25.69 | 46.28 | 12.18 | 2.3 | 1.8 |

In this section, we present the inference time of our framework when applied to the SOC method. In particular, we consider the application of a single SOC layer with various numbers of channels. Fig. 6 illustrates speedups when a single SOC layer is accelerated using the proposed method with different rank values. The figure suggests that for larger residual networks that contain layers with a number of channels up to 1000, the speedup can be up to $\approx 6$ times for $c/4$ and up to $\approx 3$ times for $c/2$. However, for networks where the number of channels is less than 500, the speedup is less than 2.

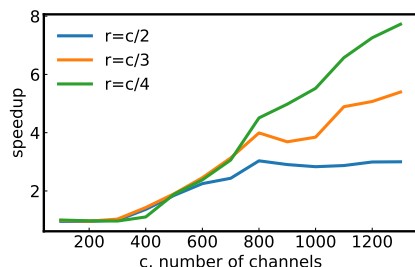

Figure 6: Speedups (w.r.t. uncompressed layer) of the application of a SOC-TT layer, $n = 16$.

### C.3 WideResNet16-10 on CIFAR-100

In addition to evaluating our framework on CIFAR-10, we conducted experiments on another classic vision dataset, CIFAR-100. We trained WideResNet-16-10 on CIFAR-100 with the same experimental setup as in Gouk et al. [2021] and compared clipping and division as in the main body of the paper. The results are presented in Tab. 4. Compared with CIFAR-10 experiments from 2, we observe that both clipping and division do not give gain on CIFAR-C (CC). The other metrics tend to improve with the control of singular values. Except for clipping-1, the TT-compressed versions with singular value control outperform the baseline accuracy in most cases. The AA metric of the baseline is outperformed when using clipping to 2 and $r = 256$.