# OpenReview forum: "Towards Practical Control of Singular Values of Convolutional Layers"
_NeurIPS.cc/2022/Conference — NeurIPS 2022 Accept_

### Official Review · Reviewer_Cy9m · 2022-07-07

**Rating:** 6
**Confidence:** 4
**Soundness:** 3 good
**Presentation:** 3 good
**Contribution:** 3 good

**Summary:**

This article proposed a framework based on the Tensor Train (TT) decomposition to decrease the complexity of computing a given convolutional layer's singular values (SVs). The key idea is to decompose the original convolution layers into low-rank factors and calculate the SVs of the factors with smaller sizes instead of the original kernel tensor. The main paper and supplementary materials give detailed proofs of the lemma and theorems. The framework is evaluated on accuracy, robustness, and calibration. The experiments are conducted on LipConv and WideResNet-28-10, using CIFAR-10 as the dataset.

**Questions:**

1. Section 2. As mentioned in the weakness part, too many topics are mentioned in this section. Though the scope of topics mentioned in this section is enormous, each of them only has short content, which makes the critical point unclear. Especially for the last two (viz. Robustness and Calibration), I think they are metrics to evaluate the framework in the experiments, thus it is not necessary to mention them in the related work section. I suggest making Section 2 more concise and the research gap more obvious.
2. About the novelty. Including TT, Tucker-2, CP, and Tensor Ring (TR) are also decomposition methods widely utilized to compress the CONV layers. I wonder why TT is employed in this work but not other decomposition methods?
3. About the choice of TT ranks. How to choose proper ranks is a vital research topic for tensor decomposition, but it is not discussed in this article. I wonder how the ranks are determined in this paper. The selection is based on experimental results? If yes, there should be an ablation study about it.
4. The experiments are insufficient. I think experiments on two models using CIFAR-10 are not enough. I think the results will be more convincing if more models (e.g., ResNet-18/50) and larger dataset (at least CIFAR-100) are considered.
5. The results lack analysis. Sections 5.1 & 5.2 are too short, and the results are listed in the Table without further discussion. It will be better if more analysis is provided. Besides, I think the first column in the right part of Table 1 should be “ACC.” instead of “AC.”? The definition of COMP. is also missing, it is not clear how to calculate it and the meaning of the number.
6. line 288 – 321. I think it is better to move this part to the appendix and use the space to add more experiments, discussion of the results, ablation study of rank, etc.


**Strengths And Weaknesses:**

Strength
1. The lemma and theorems are clearly stated, and the proofs are detailed.

Weakness
1. The originality of the proposed framework lacks significance, since using tensor decomposition to decompose a CONV layer into sequential smaller ones is not novel.
2. The related work contains too many topics, which instead blurs the key point of this article. It makes the research gap and the motivation of the proposed framework unclear.
3. This work does not give a clear strategy to find the proper TT ranks, which is of great significance for TT decomposition.
4. The experiments are not comprehensive and lack discussion and ablation study.

---

> ### Author Response · Authors · 2022-08-02
> **Response to Initial Review**
>
> We thank reviewer Cy9m for the feedback and comments.
> In addition to answers below, we invite the reviewer to check out the top-level post.
>
> > The originality of the proposed framework lacks significance, since using tensor decomposition to decompose a CONV layer into sequential smaller ones is not novel.
>
> Decomposing a kernel tensor of a convolutional layer is indeed not a novel part of our work. Our main contribution is that we find a decomposition that allows us to control singular values of the decomposed layer efficiently, and we rigorously prove the validity of our method. We provide the reasoning about significance of downstream tasks leading to the need of singular values control emphasized in the top level post.
>
>
> > The related work contains too many topics.
> > I suggest making Section 2 more concise and the research gap more obvious.
>
> We thank the reviewer for these suggestions. We will be sure to improve the related work section in accordance with your suggestions.
>
>
> > This work does not give a clear strategy to find the proper TT ranks, which is of great significance for TT decomposition.
> also
> > About the choice of TT ranks. How to choose proper ranks is a vital research topic for tensor decomposition, but it is not discussed in this article. I wonder how the ranks are determined in this paper. The selection is based on experimental results? If yes, there should be an ablation study about it.
>
> Thank you for pointing out this issue. The ranks were chosen as the maximum number of channels divided by a small constant c (such as 2-4) for each layer to control the performance by a single hyperparameter c for simplicity. Such rank policy assumes decomposing only the largest layers, and keeps the smallest layers intact. We will adjust text to make these details clear and add the ablation study w.r.t. the rank values as you suggest.
>
>
> > The experiments are insufficient and lack discussion and ablation study.
> > I think experiments on two models using CIFAR-10 are not enough. I think the results will be more convincing if more models (e.g., ResNet-18/50) and larger dataset (at least CIFAR-100) are considered.
> > The results are listed in the Table without further discussion
>
> We provide justification of the design choices and selection of evaluation protocols in the top-level post. We, however, agree that the paper will benefit from more experiments with resnet-type architectures and plan to add experiments on CIFAR-100 dataset as you suggest. Moreover, we already plan to add more discussion and ablation study as we pointed out to other reviewers.
>
>
>  In our work we focused on the study of network architectures with large convolutional layers.
>
>
> > About the novelty. Including TT, Tucker-2, CP, and Tensor Ring (TR) are also decomposition methods widely utilized to compress the CONV layers. I wonder why TT is employed in this work but not other decomposition methods?
>
> This is a great question, which we touch on in LL. 183-184 and also plan to elaborate in detail in the revised version. The short answer is that not all tensor decompositions of kernel tensors admit efficient computation of singular values. In our setting, TT serves well for this purpose, but CP and TR decompositions are incompatible by design for controlling singular values.
>
>
> > I think the first column in the right part of Table 1 should be “ACC.” instead of “AC.”? The definition of COMP. is also missing, it is not clear how to calculate it and the meaning of the number.
>
> Thank you also for pointing out issues with "ACC" and "COMP". The latter denotes the compression ratio in terms of parameters count (line 265).
>
>
> > line 288 – 321. I think it is better to move this part to the appendix and use the space to add more experiments, discussion of the results, ablation study of rank, etc.
> > Sections 5.1 & 5.2 are too short
>
> Thank you for this and other suggestions regarding clarity and readability. We will make sure to update Section 5.

---

### Official Review · Reviewer_9EwE · 2022-07-08

**Rating:** 7
**Confidence:** 3
**Soundness:** 3 good
**Presentation:** 3 good
**Contribution:** 3 good

**Summary:**

This paper proposes an efficient method to compute eigenvalues of convolutional layers based on a sparsity-driven approach. The proposed work achieves by proposing a novel framework that compresses the layer representation (section 3.2) based on a tensor decomposition. The authors provide theoretical justification in Lemma 1 and Theorem 1 of their proposed framework. In addition, empirical evidence is provided to demonstrate how effective their tensor decomposition across a wide range of architecture and metrics.

The contributions are listed as the following:
1). Proposes a novel framework Tensor-Train (TT) decomposition to compute singular values that reduce computational complexity.
2). Extend previous work on the exact computation of singular values to strided convolutions
3). Demonstrated significance of  proposed framework with varying control over the singular values on a few CNN architectures by comparing  their method to the baseline across some important metrics (accuracry, generalization error, and  adversarial robustness).

**Questions:**

1). Could you please provide some context w/ respect to Figure 1? Such as if you have speed up of x amount, could that lead to requiring less memory in GPU compute? For example, say a researcher only has access to 8 GB GPU could they compress the network by a certain factor to fit onto their GPU.

2). Strided convolutions are a large part of Generative networks in GANs, do you think your work could be built upon to study mode collapse in GAN's based on the singular value composition? As well, did you consider study these types of models in your paper?


**Limitations:**

Limitations are in section 6, assumptions are reasonable.

**Strengths And Weaknesses:**

Strengths-
1). Works on important problems to the ML/AI community.
2). Approves upon previous work by reducing the computation complexity and extending work to strided convolution.
3). Convincing empirical evidence that this framework is effective.

Weakness-
1). Theoretical claims could be better explained/justified. It's not obvious upon reading this how all the theoretical contributions are related to the work. Theorem 1 and lemma 1 are related to TT framework, and theorem 2 is about explicit singular value formula for strided convolution. To help clear some ambiguity, it might be helpful to put the lemma in appendix and expand up the technical details of theorem 1 and the importance of it for the framework. Then explain how all the theoretical contributions fit into algorithm (1).

2). I believe the paper would benefit from additional experiment, similar to the one in Figure (1), which would include speed up versus some performance metrics (accuracy, ECE). This would be very useful from a practitioner's standpoint

---

> ### Author Response · Authors · 2022-08-02
> **Response to Initial Review**
>
> We thank reviewer 9EwE for the feedback and comments.
> In addition to answers below, we invite the reviewer to check out the top-level post.
>
> > Theoretical claims could be better explained/justified. It's not obvious upon reading this how all the theoretical contributions are related to the work. Theorem 1 and lemma 1 are related to TT framework, and theorem 2 is about explicit singular value formula for strided convolution. To help clear some ambiguity, it might be helpful to put the lemma in appendix and expand up the technical details of theorem 1 and the importance of it for the framework. Then explain how all the theoretical contributions fit into algorithm (1).
>
> Indeed, Theorem 1 and Lemma 1 are related to the TT framework. In particular, Theorem 1 suggests that one can apply a method of choice to a smaller kernel K2 instead of K. Theorem 2 is focused on the generalization of exact computation of all singular values [Sedghi et al., 2019] to the case of strided convolutions, which often appear in practice. Since computation of all singular values is both time and memory consuming, our approach is particularly useful in this scenario. Anyway, we thank the reviewer for the suggestion regarding the readability. We will fix it in the next revision of the paper.
>
>
> > I believe the paper would benefit from additional experiment, similar to the one in Figure (1), which would include speed up versus some performance metrics (accuracy, ECE). This would be very useful from a practitioner's standpoint
>
> Thank you for this suggestion. We found that the choice of the rank r in terms of the number of channels m yields the best results when r = m/c, where c = 2-4 (as in Figure 1). However, we agree that the dependence of the performance on the rank could be very useful for a reader and plan to add the plot to supplementary materials.
>
>
> > Could you please provide some context w/ respect to Figure 1? Such as if you have speed up of x amount, could that lead to requiring less memory in GPU compute? For example, say a researcher only has access to 8 GB GPU could they compress the network by a certain factor to fit onto their GPU.
>
> Indeed, using our framework leads to less memory for small ranks. In particular, the gain is quadratic with respect to the ratio m/r, where m is the number of channels and r is the TT-rank. Say, if m/r=3, then our approach requires 9 times less memory. This is important for networks with large number of input/output channels. For example, if m = 4096 and n = 16 (spatial dimension), then storing n x n x m x m padded kernel tensor requires ~34GB, while our method would only require ~4GB. In the next revision, we will add a figure similar to Figure 1, but with memory requirements.
>
> > Strided convolutions are a large part of Generative networks in GANs, do you think your work could be built upon to study mode collapse in GAN's based on the singular value composition? As well, did you consider study these types of models in your paper?
>
> We believe so. We actually plan to address this matter in our future work. Similar attempts have been done, e.g., in [Liu et al., 2019], but using heuristics for computing singular values.

---

> > ### Comment · Reviewer_9EwE · 2022-08-08
> > **Comments to rebutal**
> >
> > Thank you for addressing my concerns!

---

### Official Review · Reviewer_eWcT · 2022-07-11

**Rating:** 6
**Confidence:** 3
**Soundness:** 3 good
**Presentation:** 3 good
**Contribution:** 3 good

**Summary:**

The work proposed a less expansive approach to calculating the singular values of convolutional layers. The proposed approach builds on the assumption that modern neural networks are highly overparametrized in such a way that a sparser surrogate network could achieve similar performance and the method of Tensor-Train (TT) decomposition. The work further empirically shows that a network's robustness could be improved by adding additional orthogonality constraint terms derived from the proposed approach and clipping calculated singular values.

**Questions:**

My major concerns are discussed in the Weaknesses section.

As an additional question, I'm wondering why the constraint discussed in Section 4.2 does not include a term to deal with K2 and why is the constraint useful without controlling K2.

Lastly, I would suggest expanding the clipping part discussed in Section 5.2 to a more comprehensive comparison between the proposed method and [1] (perhaps adding some comparisons when [1] is tractable in order to show both the efficiency and accuracy of the proposed method).

**Strengths And Weaknesses:**

Strengths:
The overall presentation of the work is clear. The relationships with prior arts are carefully discussed. And the experimental part clearly demonstrates the benefits of applying the proposed method and also discusses many technical details.

Weaknesses:
The reviewer found that the title of the work is not well aligned with the actual content. The title indicates that a more efficient approach for calculating singular values of convolutional layers will be proposed, one would expect to directly use the proposed method to calculate singular values of some given network (like the algorithm proposed in [1]). However, the calculation described in this work heavily relies on the decomposition (K = K1 ∘  K2 ∘  K3) and it is not clear to the reviewer how to decompose a given convolutional kernel to such form (please correct me if I'm wrong). Rather, the reviewer believes that the major accomplishment of the work lies in designing networks/constraints to improve the robustness of networks.
Besides, the experiment section seems a little bit weak to only compare with SOC only (Table 1) and the vanilla setting (Table 2). The effect of the proposed approaches can be better demonstrated If more experiments can be added.

[1] Hanie Sedghi, Vineet Gupta, and Philip M. Long. The singular values of convolutional layers, 2019

---

> ### Author Response · Authors · 2022-08-02
> **Response to Initial Review**
>
> We thank reviewer eWcT for the feedback and comments.
> In addition to answers below, we invite the reviewer to check out the top-level post.
>
> > title of the work is not well aligned with the actual content.
>
> Thank you for pointing out this concern. We agree that the title could be improved. Tentatively, "Towards practical control of singular values of convolutional layers" would reflect the merits of the proposed framework better. In case of positive decision, we will work with ACs to change the title.
>
>
> > the experiment section seems a little bit weak to only compare with SOC only (Table 1) and the vanilla setting (Table 2). The effect of the proposed approaches can be better demonstrated If more experiments can be added.
>
> In our experiments, we considered 3 principled approaches: exact computation of all singular values, direct parametrization of all singular values (SOC) and constraining the largest singular value (e.g., computing it with power iterations and subsequent division of the tensor). To the best of our knowledge, we investigated state-of-the-art methods in all categories.  In case the reviewer can suggest a reference that we could use in our study, we would be happy to consider it in the next revision.
>
>
> > As an additional question, I'm wondering why the constraint discussed in Section 4.2 does not include a term to deal with K2 and why is the constraint useful without controlling K2.
>
> The central matrix, K2, is controlled by clipping, iterative method, or SOC, while the frame matrices K1 and K3 are orthogonalized approximately with the help of auxiliary loss. According to Theorem 1, if K1 and K3 are orthogonal, the singular values of the entire convolution will match the singular values of the convolution defined by K2, which are controlled efficiently by the method of choice.
>
>
> > Lastly, I would suggest expanding the clipping part discussed in Section 5.2 to a more comprehensive comparison between the proposed method and [1] (perhaps adding some comparisons when [1] is tractable in order to show both the efficiency and accuracy of the proposed method).
>
> We appreciate the reviewer's suggestion regarding a more thorough comparison with [Sedghi et al., 2019] in the intractable scenarios, which we will add in the next revision.

---

> > ### Comment · Reviewer_eWcT · 2022-08-08
> > **Comments on Rebuttal**
> >
> > Thanks for addressing my concerns!
> >
> > I want to clarify that for the last point:
> > > We appreciate the reviewer's suggestion regarding a more thorough comparison with [Sedghi et al., 2019] in the intractable scenarios, which we will add in the next revision.
> >
> > I'm suggesting doing more comparison with [Sedghi et al., 2019] when it is tractable, not the other way around.
> >
> > Otherwise, thanks for the response!

---

### Official Review · Reviewer_EXfn · 2022-07-11

**Rating:** 6
**Confidence:** 4
**Soundness:** 3 good
**Presentation:** 3 good
**Contribution:** 3 good

**Summary:**

The authors of this paper offer a method to compute the singular values (SVs) of convolutional layers efficiently. They propose a method to decompose the matrix $\mathcal{K}$ representing a convolutional layer to smaller matrices $\mathcal{K}^{(1)} \mathcal{K}^{(2)} \mathcal{K}^{(3)}$ and show that the SVs of $\mathcal{K}$ can be expressed by the union of the SVs of some smaller matrices related to $ \mathcal{K}^{(2)}$.

**Questions:**

Please see the Weaknesses above for the questions.

**Limitations:**

The limitations are not sufficiently discussed. For example, the assumptions of the lemma and theorems are not specified clearly.

**Strengths And Weaknesses:**

## Strengths
- The decomposition of the matrix for convolutional layers is neat.
- The experimental results on clean and robust accuracy are improved with the proposed method.

## Weaknesses
- The assumptions of the lemma and theorems are not specified clearly. Do they apply to all kinds of convolutional layers?
- It seems that Theorem 2 requires $n \equiv 0\ (\text{mod}\ s)$, which means the spatial size $n$ has to be divisible by the stride $s$. Does it mean the stride equals the kernel size (i.e., $s = k$)? Many convolutional layers may not satisfy such a requirement.
- The experiments of computational complexity are too limited. A detailed comparison (in terms of, for example, running time) between the proposed method and prior methods should be added.
- The correctness of the singular values (SVs) using the proposed method should be validated empirically. For example, the SVs computed by the proposed method can be compared to those from [1]. Note that the method from [1] may only apply to convolutional layers with a stride equaling 1. For other strides, an alternative comparison is [2] by using the inequalities of matrix norms: $r\|A\|_\alpha\leq\|A\|_\beta\leq s\|A\|_\alpha$, i.e., by empirically showing that the largest singular value from the proposed method can be bounded by the $\ell_1$ norm or $\ell_\infty$ norm from [2].

[1] Sedghi H, Gupta V, Long PM. The singular values of convolutional layers. arXiv preprint arXiv:1805.10408. 2018 May 26.

[2] Liang Y, Huang D. Large norms of CNN layers do not hurt adversarial robustness. InProceedings of the AAAI Conference on Artificial Intelligence 2021 May 18 (Vol. 35, No. 10, pp. 8565-8573).

---

> ### Author Response · Authors · 2022-08-02
> **Response to Initial Review**
>
> We thank reviewer EXfn for the feedback and comments.
> In addition to answers below, we invite the reviewer to check out the top-level post.
>
> > The assumptions of the lemma and theorems are not specified clearly. Do they apply to all kinds of convolutional layers?
> > Does it mean the stride equals the kernel size? Many convolutional layers may not satisfy such a requirement.
>
>
> Thank you for the comment, we will be sure to edit the paper and write it more explicitly.
> Theorem 1 is rather general. The key assumption is the orthogonality of K1 and K3. As for the map given by K2, we only assume its linearity, which covers different convolution types.
> In Theorem 2, we follow [Sedghi et al., 2019] and assume that the convolution is periodic. As opposed to that paper, our generalization is applicable for strided convolutions (s>1). The only assumption we impose on s is that the image size n is divisible by s (or n = 0 (mod s)), which is natural for strided periodic convolutions.
>
>
> > The experiments of computational complexity are too limited. A detailed comparison (in terms of, for example, running time) between the proposed method and prior methods should be added.
>
>
> We consider 3 types of methods: using clipping [Sedghi et al., 2019], several iterations of power method [Gouk et al., 2020] and the exact parametrization [Singla and Feizi, 2021b]. Time performance for the first two methods are presented in Figure 1 and Table 2.
> To reinforce empirical study, we will add a figure similar to Figure 1 for the SOC method and present run time ablation of several new architectures. For example, our preliminary results show that clipping singular values of all layers of resnet-50 is almost 5 times slower than doing so using our framework with the rank value 256.
>
>
> > The correctness of the singular values (SVs) using the proposed method should be validated empirically.
>
>
> As correctly pointed out by the reviewer, the result from [Sedghi et al., 2019] works only for stride equal to 1. To validate correctness of our approach, we performed the following sanity check: we generated random orthogonal matrices K1, K3 and random K2, assembled a single core K from them and then constructed the full matrix of the linear map for small enough number of channel and image size to fit into computer memory. Then we computed singular values of this matrix by calling np.linalg.svd and compared it with the results obtained through our equations. All the experiments showed that they are equal up to machine epsilon. We thank the reviewer for pointing out the paper [Liang et al, 2021].

---

> > ### Comment · Reviewer_EXfn · 2022-08-10
> > **Comments to rebutal**
> >
> > I would like to thank the authors for addressing my concerns, and I raised my rating based on the responses.

---

### Author Response · Authors · 2022-08-02
**Top-level Response to Initial Reviews**

We thank the reviewers for their time, valuable comments and constructive criticism. Quoting the reviews: the work is on "important problems to the ML/AI community" (Reviewer 9EwE), the presentation is "clear" (Reviewer eWcT), the experiments part demonstrates "the benefits of applying the proposed method" (Reviewer eWcT) and the "significance of proposed framework" (Reviewer 9EwE). "The lemma and theorems are clearly stated, and the proofs are detailed" (Reviewer Cy9m).

Before giving detailed answers to the questions raised by the reviewers in the comment section, we briefly reiterate the main idea of our paper and highlight the motivation behind our numerical experiments.

Our main contribution is a framework for convolutional layer representation that allows us to reduce complexity of controlling its singular values with a method of choice (such as exact computation using FFTs [Sedghi et al., 2019], direct parametrization [Singla and Feizi, 2021b], etc.). The importance of efficiently controlling singular values stems from the practical needs typically solved through such control: the demand for certifiable robustness of deep models, stability of training, to name a few. The methods for singular values control mentioned above exhibit poor scaling with the growth of the number of convolutional layer channels. Our framework permits exploiting any of these methods in their most favorable performance regime. In our experiments, we apply our framework with these methods and demonstrate that different architectures (WideResNet, LipConvNet) benefit from using our approach.

Each method employed within our proposed framework carries over the evaluation conventions used by the authors. That is, finding the definitive set of networks, datasets, and metrics, to unify the findings, is a hard problem in its own right. On top of that, our approach to controlling singular can be used with a variety of policies (such as truncation, division, etc.). We attempted to demonstrate the most important selection of the axes of study, however, it is only natural that some important experiments were omitted. Nevertheless, we agree with comments of the reviewers that the paper would benefit from some more experiments, which we do plan to add in the revised version and which we discuss in the comment section.

---

### Meta-Review · Area_Chair_EqBb · 2022-08-26

**Recommendation:** Accept
**Confidence:** Certain

**Metareview:**

This paper introduced a tensor decomposition, and associated theory, which allows for the control of singular values in convolutional layers.

Based upon the reviews, rebuttal, and reviewer discussion, I recommend paper acceptance. All reviewers recommend acceptance. The rebuttal was effective, with one reviewer who initially recommended rejection raising their score.

The authors should be sure to follow through, and update the paper to include changes discussed during the review period. Especially, it seems as if the framing of the paper shifted during the review period from centering on practically computing singular values to practically controlling singular values. From my understanding of the work, I agree this second framing makes more sense.

**Award:**

No

---

### Decision · Program_Chairs · 2022-09-14

Accept